

# Effects of changes in climatic conditions on soil water storage patterns

Annelie Ehrhardt[1,2], Jannis Groh[2,3,4], Horst H. Gerke[5]

[1]TU Bergakademie Freiberg, Institute for Drilling Technology and Fluid Mining, Agricolastraße 22, 09599 Freiberg

[2]Leibniz Centre for Agricultural Landscape Research (ZALF), Research Area 1 "Landscape Functioning", Working group "Isotope Biogeochemistry and Gas Fluxes", Eberswalder Straße 84, 15374 Müncheberg, Germany

[3]University of Bonn, Institute of Crop Science and Resource Conservation (INRES) -- Soil Science and Soil Ecology, Nußallee 13, 53115 Bonn

[4]Institute of Bio- and Geoscience IBG-3: Agrosphere, Forschungszentrum Jülich GmbH, Jülich, 52425, Germany

[5]Leibniz Centre for Agricultural Landscape Research (ZALF), Research Area 1 "Landscape Functioning", Working group "Silicon Biogeochemistry", Eberswalder Straße 84, 15374 Müncheberg, Germany (https://orcid.org/0000-0002-6232-7688)

*Correspondence to*: Annelie Ehrhardt (annelie.ehrhardt@zalf.de)



**Abstract.**

The soil water storage (SWS) defines crop productivity of a soil and varies under differing climatic conditions. Pattern identification and quantification of these variations remains difficult due to the non-linear behaviour of SWS changes over time.

We hypothesize that these patterns can be revealed by applying wavelet analysis to an eight-year time series of SWS, precipitation (P) and actual evapotranspiration (ETa) in similar soils of lysimeters in a colder and drier location and a warmer and wetter location within Germany. Correlations between SWS, P and ETa at these sites might reveal the influence of altered climatic conditions but also from subsequent wet and dry years on SWS changes.

We found that wet and dry years exerted influence on SWS changes by leading to faster or slower response times of SWS changes to precipitation in respect to normal years. Extreme precipitation events were visible in SWS and P wavelet spectra. Time shifts in correlations between ETa and SWS became smaller at the wetter and warmer site over time in comparison to the cooler and drier site where they stayed constant. This could be attributed to an earlier onset of the vegetation period over the years and thus to an earlier ETa peak every year and reflects the direct

impact of changing climate on soil water budget parameters.

Long-term observations (>30 years) might reveal similar time shifts for a drier climate. Analysis of the SWS capacity could provide information on how different climatic conditions affect the long-term storage behaviour of soils.

# 1 Introduction

The soil water storage capacity (SWSC) is defined as the amount of water stored within the plant root accessible upper part of the vadose zone (e.g., Kutilek & Nielsen, 1994). Both, the SWSC and the process of soil water storage (SWS) within the root zone are important for defining the crop productivity (e.g., Stocker et al., 2023). The SWS in the vadose zone, i.e., the region between surface and groundwater table, has furthermore been considered a key for understanding ecohydrological interactions within the soil-

water-atmosphere continuum (Vereecken et al., 2022).

The SWS is a dynamic component of the soil or ecosystem water balance equation and varies within the usually assumed constant SWSC. The SWS is the residual between in-flux and out-flux components of the soil water balance; SWS has also been determined in the field by vertically integrating the soil water





content obtained by point measurements using either soil moisture sensors or soil samples (gravimetric
method) (e.g., Kutilek & Nielsen, 1994). Observation methods for quantification of the soil water balance
for larger soil volumes include lysimeters, hydro-gravimeters, or cosmic-ray neutron sensor networks
(Heistermann et al., 2022). The SWS increases due to infiltration by rainfall, irrigation, non-rainfall
precipitation (e.g., Groh et al., 2018), or upward-directed water movement from deeper soil layer or
groundwater and lateral subsurface flow at hillslopes (e.g., Rieckh et al., 2014). The SWS decreases due
to actual evapotranspiration (ETa), lateral outflow, or vertical drainage. Annual changes in SWS have
been used to quantify impacts of climate variability on plant growth and crop production (He & Wang,
2019) or to analyze the susceptibility of soils towards floods and droughts (Shah & Mishra, 2021). The
analysis of SWS changes was used to explain effects soil moisture variability on nutrient (Li et al., 2010,
Shen et al., 2022) or carbon cycling (Lal, 2019) and ETa in different land-use systems (Yang et al., 2016,
Rahmati et al., 2020). The SWS depends on soil texture (e.g., Tafasca et al., 2020), soil structure (e.g.,
Rabot et al., 2018), organic carbon content (e.g., Hu et al., 2017), and vegetation properties (Trautmann
et al., 2022). Recent studies have shown that reoccurring drought years since 2015 have left severe deficits
in the total water storage of catchments (Laaha et al., 2017) and continents (Boergens et al., 2020) that
are unprecedented in the past 2110 years (Büntgen et al., 2021). Groh et al. (2020) found that droughts
can have an impact on the long-term SWS. The observation showed that SWS declined after a drought in
2015 and remained depleted until the end of the observation period, which implies long-term effects of
droughts (e.g., on the SWSC) and more importantly, the carry-over of the drought from one growing
season to the next one. However, the SWS dynamics and their feedback to climate systems have been
considered difficult to observe and comprehend (Vereecken et al., 2022, Groh et al., 2020, Herbrich &
65  Gerke, 2017).

A common concept is that the SWS dynamics in the northern temperate climate zones have a dominant
annual cycle (Stahl & McColl, 2022) with the decrease during the growing period (ETa>P) and the
increase during the non-growing winter period (P>ETa). In the longer term, the SWS approaches a soil-
and site-specific mean value, which is usually defined according to the situation in the late spring (Groh
70  et al., 2020) just before the beginning of the growing period. The soil moisture conditions at this time of
the year can be assumed to be optimally rewetted and in hydrostatic equilibrium. Water balance





calculations are mostly assuming that the SWS approaches approximately the same value at field capacity in late spring and that the SWSC remains constant.

Of course, the SWS patterns may differ within the annual cycles for agricultural crops and natural vegetation (Jia et al., 2013). Longer-term changes in SWS patterns and SWSC can be expected when the soil properties are changing, which has been reported from situations of soil degradation and amelioration, changes in land use and soil management (e.g., Palese et al., 2014; Yu et al., 2015). However, the effect of a change in climatic conditions on SWS has scarcely been reported to date. Robinson et al. (2016) demonstrated a drought induced alteration of soil hydraulic properties and a decrease in SWS, but only indirectly using soil moisture observations that are not representative for the effective root zone but rather a small fraction of the soil. This lack of studies results from methodical difficulties in determining dynamic changes because of the complex effects that account for changes in SWS at shorter and longer time scales (Chen et al., 2023).

To analyse these dynamics and derive reoccurring patterns in time series of SWS, a variety of methods including principal component analysis (PCA), empirical orthogonal functions (EOF), wavelet transform, unsupervised learning like self-organizing maps (SOM), empirical mode decomposition (EMD) have been applied (Vereecken et al., 2016). Hohenbrink et al. (2016) used PCA to explain influences of meteorological boundary conditions and different cropping systems on soil water dynamics. Korres et al. (2010) employed EOF to identify differences in governing factors on soil moisture variability between a grassland and a cropland soil and found that this method was not only suitable to detect temporally stable patterns caused by soil parameters but also to find non-stable patterns evoked by different land management practises. The SOM method was found useful to derive vertical and lateral flow events along surface or subsurface boundaries from a time series of soil water content data (Lee & Kim, 2021). To analyse the scale specific time stability of SWS, Hu et al. (2014) showed with multivariate EMD that the time stability of the SWS signal was strong within same seasons but weak between different seasons. The mentioned methods for time series analysis are efficient for revealing overall patterns that occur throughout the signal. However, these approaches do not allow to localize these patterns in time as it could be done with a wavelet analysis. Especially, it is not possible to determine whether annual or daily cycles within a signal are occurring over the entire period or if these patterns are interrupted in time.





The wavelet analysis decomposes a time series into several components each accounting for a certain frequency band by comparing the signal with a set of wavelet functions of known frequency. Therefore the convolution integral is calculated between the signal and a series of scaled wavelet functions derived from the same mother wavelet (Farge, 1992). That is similar to Fourier transform that derives the dominant frequency of a signal from calculating the convolution integral between a time series and a set

of sinusoidal functions. However, since the wavelet function has zero mean, it is localized in time. That means at every moment from a time series the dominant frequencies can be derived with wavelet analysis in contrast to the Fourier analysis that calculates only the dominant frequency across the entire time series. Additionally, two time series can be correlated by wavelet coherency analysis (WCA) revealing the similarity between two signals that might have been overlooked by traditional correlation analysis

(Grinsted et al., 2004). For example, if two time series contain similar frequencies but are only shifted in time against each other Pearson correlation indicates only little similarity between the signals in contrast to WCA (Bravo et al., 2019).

wavelet coherence analysis has been applied to reveal different temporal correlations between matric potential and precipitation for grassland and cropland (Yang et al., 2016). Liu et al (2017) showed that a

change in water uptake strategies between grassland and woodland is manifested in a decreasing correlation between soil moisture and precipitation at high frequencies. Graf et al. (2014) investigated the spatiotemporal relations in a forested catchment located in West Germany between water budget components and soil water content. The WCA was used here to identify the main source of uncertainty when closing water balance at smaller time scale (daily, weekly etc.). With WCA it is not only possible

to derive correlations across different scales from non-linear data like SWC or ETa time series but also the temporal shifts between those reoccurring patterns (e.g., Rahmati et al., 2020). To derive differences in temporal onset between a lysimeter and a field soil that might indicate the occurrence of lateral subsurface flow in hummocky landscapes, Ehrhardt et al. (2021) applied WCA to a cropland soil. They attributed the faster SWC increase in the field soil in comparison to the lysimeter soil to water entering

the field soil laterally from higher terrain positions. Ding et al. (2013) showed that after irrigation pulses the time shift between ETa and SWC changed at daily scale demonstrating the control of irrigation on the temporal variability of ETa.



When analysing the effect of climate variability on SWS it is plausible to compare time series of similar soils under different climatic conditions (i.e., space-for-time substitution approach, e.g., Groh et al., 2020). However, as the soil develops differently under each local climate, the same soil can hardly be found under a different climate. The situation can only be created experimentally. Within the TERENO-SOILCan lysimeter-network (TERrestrial ENvironmental Observatories; Pütz et al., 2016) lysimeters extracted from different land use types (natural and managed grassland, arable land) and soil types are transferred according to a modified space-for-time approach to sites with differing climatic conditions. This setup allows to evaluate the impact of altered climatic conditions on agricultural ecosystems (Pütz et al., 2016) and to quantify changes in the soil water cycle and crop production caused due to climate variability. In previous studies, the soil water balance components of the lysimeter at the original location have been compared with those of the lysimeter transferred ones to define the impact of changing climate and management on nitrogen leaching (Fu et al., 2017), to evaluate precipitation measurement methods (Schnepper et al., 2023), and modelling hydrological processes and ecosystem productivity of the same soil but under different climate for arable-land and grassland ecosystems (Jarvis et al., 2022, Groh et al., 2023). Rahmati et al. (2020) demonstrated that due to increasing dryness, the SWS is stronger controlled by ETa for a grassland soil. They explained declining phase shifts between ETa and SWS at the annual scale over a 7-years period with increasing dryness and suggested that this might also be the case for cropland soil.

Still, in cropland long-term studies on trend analysis and pattern detection in SWS time series to derive the effect of changing climate on SWS components are limited and restricted to larger scales like satellite observations (e.g., GRACE-REC, Humphrey & Gudmundsson, 2019). Agboma and Itensifu (2020) observed increasing periodicity in SWS changes with increasing soil depth that might be relevant for seasonal soil moisture regime forecasting. They concluded that such studies are still missing for cultivated cropland because most monitoring sites for SWS observation are in grassland. Chen et al. (2023) identified different governing parameters on SWS stability in winter and summer highlighting the need for these analyses on long-term data to derive impacts of extreme climate change on hydrological variables.





To gain more insights on SWS patterns evolving under differing climate conditions in cropland for an eight-year observation period (2014 until 2021) we employ WCA to compare SWS time series of a soil at its original location to this soil transferred to a wetter and warmer climate. We hypothesize that similar to grassland soils the phase shift between ETa and SWS is smaller under drier as compared to wetter conditions. Our objectives are (i) to detect temporal patterns in SWS changes (ΔSWS) of the same soil

under two different climatic conditions (drier and colder vs. wetter and warmer) with wavelet analysis and (ii) to visualize how other soil water balance components (precipitation P, ETa, net drainage) are affected or affect the ΔSWS under different climatic conditions. We expect a quantitative temporal off-set between daily, seasonal, and annual changes in the components of the soil water balance and effects on SWS patterns from those of wet climatic conditions (2015-2017) to change in subsequent dry years

165 (2018-2020).

## 2 Materials and Methods

### 2.1 Site description

The study areas are located in Selhausen (51°52'7''N, 6°26'58''E) and Dedelow (53°23'2''N, 13°47'11''E) (Fig. 1). A total of nine high-precision weighing lysimeters (precision: 10 g, METER

Group) were filled with intact eroded Luvisol soil monoliths in Dedelow. Three out of nine lysimeters were installed in Dedelow and three were transferred to Selhausen, and Bad Lauchstädt each to expose the extracted soil to different climate regimes. For the purposes of this study, we will only address the lysimeter measurements at the Dedelow and Selhausen sites. The transfer from Dedelow to Selhausen corresponds to an increase in annual precipitation sum of 112 mm and an increase in average annual

temperature of 1.6 °C throughout the study period (2014-2021).

The experimental set-up is part of the TERENO-SOILCan lysimeter network (Pütz et al., 2016). The lysimeters are 1.5 m deep and have a surface area of 1 m². The soil water dynamics at the lysimeter bottom were adjusted to field conditions by a bi-directional pumping control system that adjusts measured pressure head at the bottom of the lysimeter to measured pressure head in a similar depth in the field.

During drainage periods water from the lysimeter was collected via a suction rake at the bottom of the



lysimeter in a weighable seepage tank (precision: 1 g). In periods with an upward directed water flow from capillary rise, the water was pumped back into the lysimeter from the seepage tank. For more details on the lysimeter set-up and equipment refer to Groh et al. (2020). The lysimeters were embedded within larger fields in Selhausen (0.025 ha) and Dedelow (2.3 ha), where the plant management in the lysimeter
and the field was identical during the observation period.

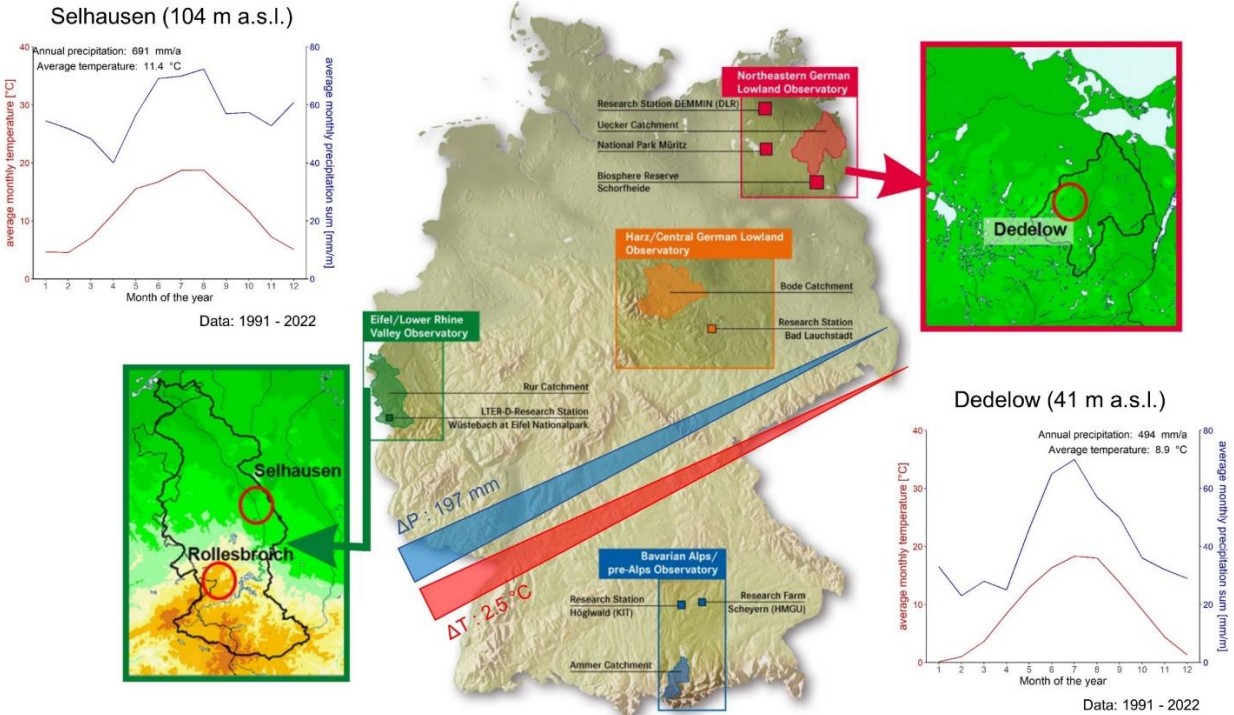

**Figure 1: Average monthly precipitation (P) sums and average monthly temperature in Selhausen and in Dedelow (between 1991 and 2022).**

The climate in Dedelow with an average annual P sum of 494 mm and an average annual temperature of
8.9 °C (1991-2022) is more continental than the climate in Selhausen with an average annual P sum of 691 mm and an average annual temperature of 11.4 °C (1991-2022). The rainfall distribution is unimodal at both sites with a peak in summer. Average monthly temperatures in Dedelow experience a minimum in January with 0.1 °C and a maximum in July with 18.3 °C, whereas temperatures in Selhausen vary between 4.5 °C in February and 18.8 °C in August indicating a slightly smaller annual temperature
amplitude between winter and summer for Selhausen. Average monthly temperatures and precipitation (Fig. 1) were obtained from automated weather stations in Dedelow (SYNMET/LOG, LAMBRECHT



meteo GmbH) and Selhausen (weather station of the Forschungszentrum Jülich; the data are available at:

https://teodoor.icg.kfa-juelich.de/ibg3searchportal2/index.jsp, station ID RU_K_011).

All soils are Haplic Luvisols. The soil monoliths were extracted at a midslope positions along a 20 m

transect of an agricultural field site as close as possible to each other (~3 m apart; Herbrich & Gerke,

2017). The texture of the Ap-, E+Bt-, and elCv-horizons was described as loamy sand. The clay content

in the Bt-horizon is slightly higher than in the other horizons indicating a more loamy texture Tab. 1).

**Table 1: Horizon depths, soil bulk density (ρ$_b$), porosity (ε) and texture (sand: 2.0 to 0.063 mm; silt: 0.063 to 0.002 mm; clay: < 0.002 mm) for the lysimeter Dd_1 located in Dedelow. The other lysimeters differed only in the thickness of the diagnostic horizons**
**below the Ap-horizons. Data are from Herbrich & Gerke (2017). Supporting information: see also Groh et al. (2022)**

| Horizon* | Depth [cm] | ρb [g cm⁻³] | ε [cm³ cm⁻³] | Sand [g kg-1] | Silt [g kg-1] | Clay [g kg-1] |
|---|---|---|---|---|---|---|
| Ap | 0-30 | 1.53 | 0.42 | 538 | 305 | 157 |
| E+Bt | 30-42 | 1.65 | 0.38 | 510 | 341 | 149 |
| Bt | 42-80 | 1.52 | 0.43 | 507 | 299 | 194 |
| elCv | 80-150 | 1.69 | 0.36 | 589 | 293 | 118 |

*Horizons named according to FAO classification (IUSS Working Group WRB, 2015)

Cover crops varied each year (Tab. 2) but were similar for Dedelow and Selhausen despite for 2014,

when oat was grown in Selhausen and Persian clover in Dedelow.

**Table 2: Cover crops, dates of sowing and harvest (format [dd-mm-yyyy]), duration of vegetation period in days (Veg. per.) and**
**amount of precipitation (P) in mm during the vegetation period for the lysimeters in Selhausen and Dedelow (average values from three repetitions)**

| | Selhausen | | | | | Dedelow | | | | |
|---|---|---|---|---|---|---|---|---|---|---|
| Year | Crop | Sowing | Harvest | Veg per. [d] | P [mm] | Crop | Sowing | Harvest | Veg per.[d] | P [mm] |
| 2014 | Oat | 05-03-2014 | 03-06-2014 | 90 | 141 | Persian clover | 04-03-2014 | 24-07-2014 | 142 | 247 |
| 2015 | Winter wheat | 15-10-2014 | 21-07-2015 | 279 | 501 | Winter wheat | 17-09-2014 | 23-07-2015 | 309 | 443 |
| 2016 | Winter barley | 07-10-2015 | 08-07-2016 | 275 | 632 | Winter barley | 02-10-2015 | 27-06-2016 | 269 | 427 |
| 2017 | Winter rye | 11-10-2016 | 21-07-2017 | 283 | 453 | Winter rye | 06-10-2016 | 02-08-2017 | 300 | 732 |
| 2018 | | | | | | Winter barley | 20-10-2017 | 11-04-2018 | 173 | 313 |
| | Oat | 15-03-2018 | 24-07-2018 | 131 | 176 | Oat | 11-04-2018 | 27-07-2018 | 107 | 101 |
| 2019 | Winter Wheat | 05-11-2018 | 24-07-2019 | 261 | 441 | Winter Wheat | 09-10-2018 | 25-07-2019 | 289 | 421 |
| 2020 | Winter barley | 30-09-2019 | 07-07-2020 | 281 | 553 | Winter barley | 26-09-2019 | 02-07-2020 | 280 | 381 |
| 2021 | Winter rye | 20-10-2020 | 04-08-2021 | 288 | 643 | Winter rye | 06-10-2020 | 26-07-2021 | 293 | 578 |

## 2.2 Soil water storage, actual evapotranspiration and precipitation data

Weight changes (i.e., the changes in mass) of the lysimeters were collected in 1-min resolution and

aggregated to hourly values. The raw data were checked manually as well as automatically according to

Pütz et al. (2016) and Schneider et al. (2021). To further reduce the impact of noise on the determination



of ETa and P data, the adaptive window and threshold filter (AWAT, Peters et al., 2017) was applied. Missing data were gap-filled on aggregated hourly basis within the post-processing scheme. In a first step, a linear regression model was applied that was using the mean value of ETa and P calculated from values

of all available lysimeters with the corresponding soil. In a second step, remaining gaps were gap-filled by a linear regression model that was using reference data from a rain gauge or reference evapotranspiration (grass) according to the Penman Monteith method (Allen et al., 1998).

Values of hourly soil water storage changes $\Delta SWS$ [mm h$^{-1}$] were calculated according to:

$$\Delta SWS = P - ET_a - Q_{net} \qquad (1)$$

where $Q_{net}$ refers here to the hourly sum of net water flux [mm h$^{-1}$] across the lysimeter bottom ($Q_{net} > 0$: drainage, $Q_{net} < 0$: capillary rise).

The cumulative change in total soil water storage, $SWS_t$ [mm], from the value at the beginning of the measurements, $SWS_0$ [mm], was obtained by integrating (i.e., which is here identical with summing hourly values) $\Delta SWS$ as:

$$SWS_t = SWS_0 + \sum_{i=1}^{N} \Delta SWS_i \, \Delta t_i \qquad (2)$$

for every hour, $i$, till the end (N = 70080 h) and for each of the lysimeters.

## 2.3 Wavelet analysis and wavelet coherence analysis

The complex Morlet wavelet (wavenumber $k_0 = 6$) was selected as a mother wavelet for the continuous wavelet transform of the time series. The Morlet wavelet is well suited for the analysis of environmental

signal due to its good balance of time and frequency resolution (Grinsted et al., 2004). Also, due to its complex nature amplitude and frequency of the signal can be reproduced (Torrence & Compo, 1998). As a background spectrum, a first-order autoregressive process (red noise) was chosen to test the significance of the wavelet spectra. For the visualization of the wavelet spectra and the wavelet coherence spectra, a significance level of 10 % against this background spectrum was applied. 300 Monte Carlo simulations

were conducted to find the regions of significant periodicities. For smoothing of the wavelet spectra, a Blackman window was selected to amplify the significance within the single wavelet spectra (Torrence and Compo, 1998). For the time series no detrending was performed. Calculation of the wavelet plots and wavelet coherence plots was performed according to Torrence & Webster (1999) and executed in the R





software v. 3.6.2 (R Core team, 2019) with the package WaveletComp (Roesch & Schmidbauer, 2018).

Variables used for the WCA were the SWS, P, ETa, and $Q_{net}$ in Dedelow and Selhausen. For correlations between the two locations the data set from Dedelow was the base signal and data from Selhausen as the second signal. P and ETa were used as the base signals for correlations between P and SWS and between ETa and SWS.

WCA does not only derive times and scales of correlation between two signals but also how the periodic

fluctuations of the time series are shifted in time against each other. General trends in phase shifts are indicated by the arrows within the significant parts of wavelet coherence spectra. They can be quantified by analysing the phase angle derived from the imaginary and real part of the cross-wavelet spectrum (Si, 2008). The phase angle is calculated in radians in the range from $-\pi$ to $+\pi$. Depending on the scale of interest, $\pi$ corresponds to a time shift of 12 h at the daily (24 h) scale and to 4380 h at the annual (8760 h)

scale.

For more details on the theoretical background of wavelet and wavelet coherence analysis refer to Si & Zeleke (2005) and Grinsted et al. (2004).

## 3 Results and Discussion

### 3.1 Comparison of SWS, ETa and P under different climatic conditions

Throughout the observation period (2014-2021), the $\Delta$SWS values ranged from -100 to +100 mm relative to the initial value of $SWS_0$ at the beginning of the period for Dedelow and between -300 and +25 mm for Selhausen (Fig. 2). The annual fluctuations in SWS were more pronounced in Selhausen (wetter and warmer climate) as compared to Dedelow (drier and colder climate). For Selhausen, the year 2015 brought an extreme decline in SWS (-300 mm) due to a drought that spread not only to the local region but to

large parts of Europe (Ionita et al., 2017). For Dedelow, the years 2018 and 2019, which included the extreme drought in 2018 (Büntgen et al., 2021), were characterized by extremely dry conditions, which led to a decrease in SWS and an early ripening of the oat crop (Groh et al., 2019).

Wetter years with a more than average P amount were 2014 (+37% above average) and 2017 (+77%) for Dedelow and 2014 (+26%) for Selhausen (Table A1). From 2014 to 2021, the total amount of P per year



decreased with minimum values of 400 mm a$^{-1}$ in 2018 for Dedelow and 534 mm a$^{-1}$ in 2018 for Selhausen. Note that the average value of P (2014-2021) was significantly higher than the P for the respective reference period (1991-2022) determined by standard rainfall gauges, which underestimate P as compared to the more realistic lysimeter P data (Schnepper et al., 2023). In addition, P amounts determined with lysimeters include water from non-rainfall events (i.e., dew formation), which

contributed 7.2 % on the annual scale of total P for the period 2015-2018, at least for Selhausen and the nearby Eifel region (Forstner et al., 2021, Groh et al., 2019).

Note the extreme increase in SWS in both locations in July 2021 that was caused by an extreme precipitation event with up to 174 mm in Dedelow and 103 mm in Selhausen within two days causing major flooding within the Eifel-Ardennes Mountains in Germany (Lehmkuhl et al., 2022).

Daily ETa rates experienced annual cycles with a maximum in 2015 for Selhausen (691 mm a$^{-1}$) and in 2017 for Dedelow (700 mm a$^{-1}$), which was 22 % and 24 % more than the average annual ETa value at the corresponding site (Table A1). The bottom drainage of the lysimeters was much smaller in Dedelow than in Selhausen (Fig. 2) corresponding to the drier climatic conditions at the more continental experimental site.



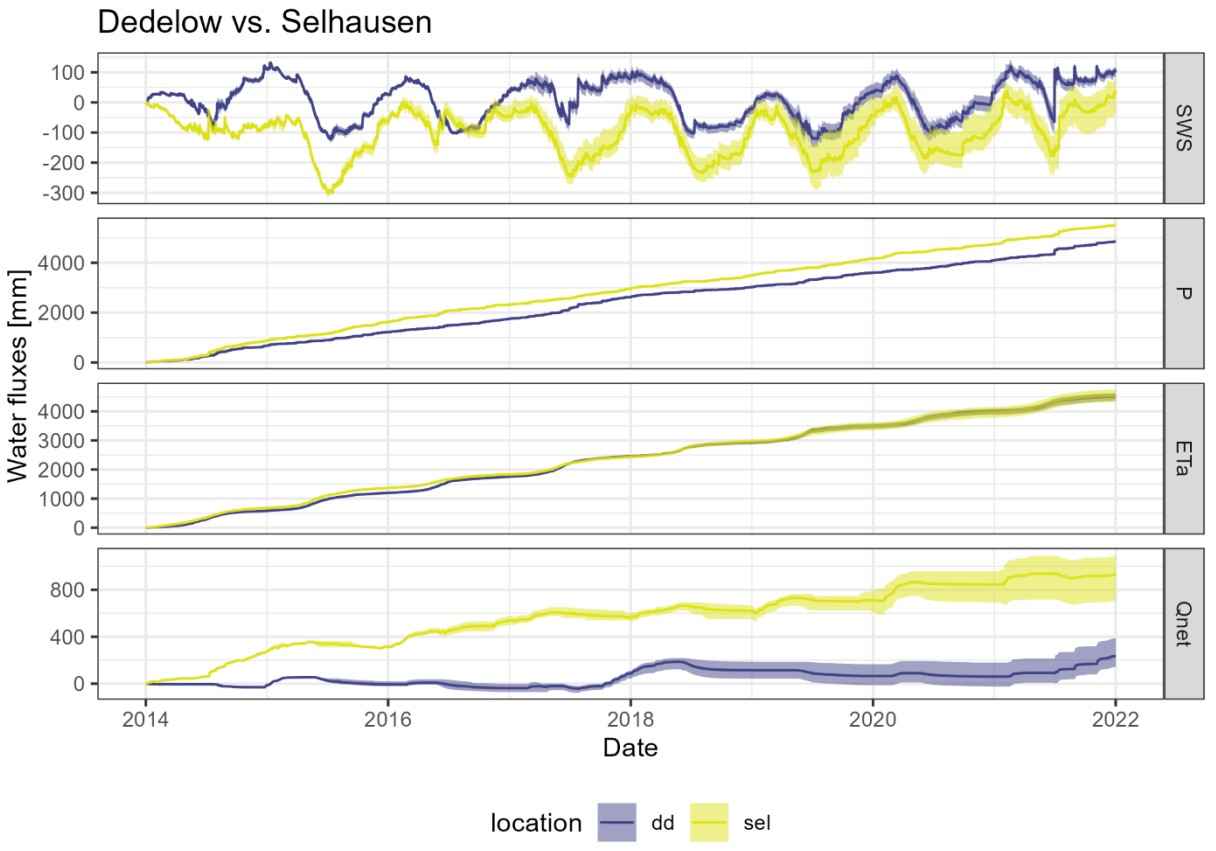

**Figure 2: Hourly soil water storage change (ΔSWS) and cumulative sum of precipitation (P), actual evapotranspiration (ETa) and bottom flux (up- and downwards, Qnet) of the lysimeters since 01-01-2014 in Dedelow and Selhausen. Shaded areas represent the cumulative minimum and maximum values of the hourly data**

These annual variations in ΔSWS and ETa observed in the time series were reflected in the wavelet spectra (Fig. 3). For SWS both wavelet spectra in Selhausen and Dedelow showed significant periodicities (area within the white edging) at the annual scale (period = 8760 h) over the entire observation period (Fig. 3 a, b). Such annual patterns in SWS changes have been also found by Liu et al. (2020) for the Shale Hills catchment in Pennsylvania; USA. They related these fluctuations to seasonal variations due to water consumption by plants (transpiration) and soil evaporation. At the daily scale (period = 24 h), a diurnal variation throughout the vegetation period was vaguely perceptible as indicated by the green band at the 24 h-scale (Fig. 3 a and b). This diurnal fluctuation was however not significant against the red-noise background spectrum.





The influence of wet and dry years was visible in the wavelet spectra of the SWS changes (Fig. 3 a, b):
At scales higher than the annual scale, significant periodicities were found for Dedelow between 2017
and 2021 and for Selhausen between 2015 and 2018 at the two-year scale. Significant periodicities in
SWS changes extended towards smaller scales in Dedelow in 2015, 2017 and 2021 that correspond to
years with more than average P (Table A1) that is also visible in the wavelet spectra of the P (Fig. 3 c).
For Selhausen, significant periodicities were found at smaller scales in the years 2014 and 2021
corresponding to years with an increased P amount (Table A1) like in Dedelow. In the wavelet spectra of
P for Selhausen, periodicities extending to smaller scales were found also for the year 2016 that has been
characterized with an extremely low ETa (-18% than average year). Also, in 2016 Selhausen received
200 mm more P throughout the vegetation period (Tab. 2), possibly explaining the differing ΔSWS
patterns in comparison to those for Dedelow.

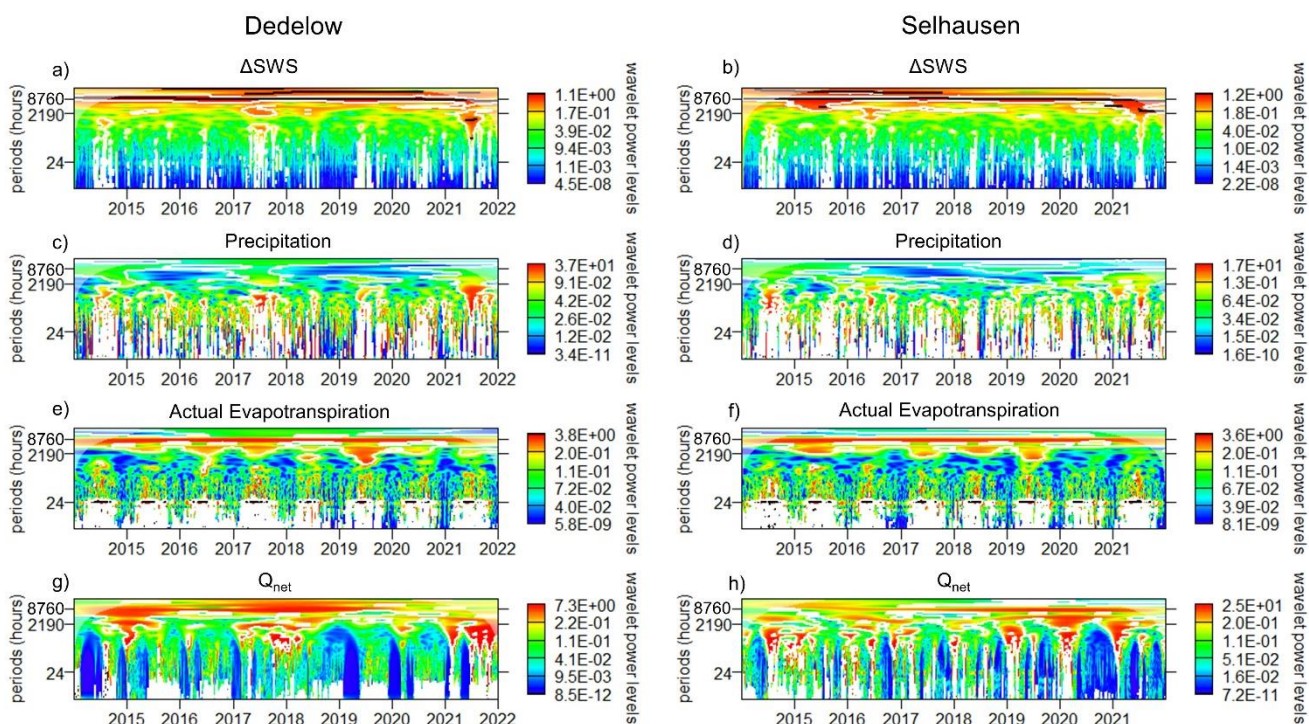

**Figure 3: Wavelet spectra of the soil water storage change (ΔSWS), precipitation, actual evapotranspiration and bottom drainage
Qnet (up- and downward flux) of the lysimeters in Dedelow and Selhausen. Time is depicted on the x-axis and the y-axis denotes the
periodicity in hours (24h = daily scale, 8760 h = annual scale). The colour indicates the wavelet power level that indicates the
similarity of the frequency of the wavelet with the frequency of the time series at the given scale and at the point in time. Areas in
the wavelet spectrum that deviate significantly from the red noise background spectrum (significance level = 10 %) are surrounded
by the white edging. A logarithmic scale for the wavelet power levels was chosen to amplify differences in wavelet coefficients between**



**different parts of the spectra visually. The shaded area at the edge of the plot at higher scales is called the cone of influence. Here, edge effects due to the padding of the time series with zeroes at the beginning and the end might influence the appearance of the wavelet spectrum and thus should be interpreted with caution.**

As already shown for Dedelow, the years 2014 and 2021 with increased P were also visible in the
significant areas of the wavelet spectrum for Selhausen (Fig. 3 d).

Extreme drought events and vegetation periods are reflected in the wavelet spectra for ETa Dedelow and Selhausen that showed distinct annual cycles (Fig. 3 e and f). Also, the periodicities at the daily scale were significant throughout the vegetation period at both sites indicating the influence of vegetation on increased ETa. In 2019 the spectra of both sites showed significant periodicities extending to smaller
scales corresponding to a year with extreme drought in Germany (Boeing et al., 2022).

The wavelet spectra of $Q_{net}$ of the lysimeters showed a distinct annual fluctuation in Dedelow from 2014 to 2019, whereas in Selhausen this annual cycle occurred between 2016 and 2021 (Fig. 3 g, h). At the drier site in Dedelow, the years with more P were well distinguishable by significant periodicities extending towards smaller scales (Fig. 3 g). In Selhausen these patterns were observed almost every year
(Fig. 3 h).

The higher amplitude in the annual fluctuations of SWS in Selhausen (Fig. 2: ΔSWS: 300 mm) in comparison to Dedelow (Fig. 2: ΔSWS: 200 mm) was reflected in the global wavelet power that is obtained when averaging the wavelet coefficients of a time series over an entire scale (Fig. 4 a, d).



**Figure 4: Global wavelet coefficients of soil water storage (a), precipitation (b), actual evapotranspiration (c) across different scales in Dedelow and Selhausen. (d) and (e) are close-ups of the SWS and ETa at the annual scale, respectively.**

This could be attributed to the higher annual P amount in Selhausen in contrast to Dedelow, especially since the ETa was similar for both sites on average (Table A1). In contrast to Dedelow, a small peak

around a period of approximately 16500 hours was found in Selhausen (Fig. 4 b) indicating a two-year cycle that was already found in the wavelet spectra (Fig. 3 d). For ETa strong peaks in the global wavelet spectra were found at the daily and at the annual scale (Fig. 4 c, e). Also, a small peak at a periodicity of around 4380 h was observed for ETa responding to a semi-annual cycle attributed to the length of the vegetation period. Note that the peaks in ΔSWS changes and ETa values around the annual scale were

occurring slightly below a periodicity of 8760 h that corresponded to the exact number of hours per year. This could indicate a temporal shifting of the annual cycles, possibly caused by a change in climatic



conditions. For example, Rahmati et al. (2023) showed that in Europe since 1981 the start of the vegetation period and the dry period was shifted towards earlier times in the year. Thus, the total difference in days between the start of the vegetation period of the preceding year and the following year decreases over time leading to a shift of annual cycles towards lower periodicities.

## 3.2 Correlation and time shifts between soil water budget variables of both sites reflect dominant climatic patterns

Correlating the SWS, P and ETa fluctuations between Dedelow and Selhausen by WCA might reveal the effects of changing climatic conditions on the soil water budget that was not directly visible from the time series itself (e.g. Biswas & Si, 2011).

Carry-over effects of dry years are found in the WCA spectra when correlating SWS changes from the drier and colder site with those from the wetter and warmer site. The coherence plot of SWS between Dedelow and Selhausen revealed a highly significant correlation pattern at the annual scale, which is only interrupted in 2017 (Fig. 5). The year 2017 has been denoted as an extreme wet year in Dedelow with almost 77 % more P than average (1991-2022). On the two-year scale, significant correlations between the two experimental sites were found from 2020 to 2021 with a positive phase shift indicating an earlier rewetting phase in Dedelow than in Selhausen (Fig. 5 b,i). This trend is opposite to the phase shifts found at the annual (Fig. 5 b,ii) and semi-annual scale (Fig. 5 b,iii). It could indicate the carry-over effect of SWS deficit from the previous drought year 2020 as already described by Groh et al. (2020) for different soils at the experimental site in Bad Lauchstädt.



**Figure 5: Wavelet coherence plots showing the correlation between Dedelow and Selhausen of SWS (a), precipitation (c) and evapotranspiration (e). For explanation of the plot layout refer to Fig. 3. The black arrows indicate the phase shift in the correlation of these variables between Dedelow and Selhausen. Arrows showing to the right indicate a perfect correlation without any shift in time. Arrows pointing upwards indicate a leading pattern for the plots in Dedelow whereas arrows pointing downwards show a leading pattern for Selhausen. These phase shifts can be expressed quantitatively in hours or days (b, d, f) for a given scale within in significant parts of the WCA spectrum. Negative and positive phase shifts correspond to a leading pattern for Dedelow and Selhausen, respectively.**



Significant correlations extended towards smaller scales (semi-annual and quarterly scales) in spring 2015, autumn 2017 and 2018, winter 2019/2020 and spring 2021 possibly reflecting the influence of plant
growth on SWS (Fig. 5a). In 2016 no correlations between the two sites were found that might be attributed to the much smaller P amount throughout the vegetation period in Dedelow compared to Selhausen (Tab. 2).

The influence of wet and dry years was reflected in changing phase shifts between the two sites. No considerable temporal deviations in SWS changes at the annual scale were found between Dedelow and
Selhausen in Fig. 5 a (arrows indicting phase shift). However, when directly plotting the phase shift from the significant parts of the WCA spectrum, a slightly negative off-set was found until 2017 at the annual and semi-annual scale (Fig. 5 b) for the variable SWS change. This refers to an in general faster decrease in SWS in Selhausen than in Dedelow. In 2017 this trend was reverted into a positive phase shift showing a faster change in SWS in Dedelow than Selhausen, due to the exceptional high P during this year at
Dedelow. After the drought year 2020, again negative phase shifts were observed at the annual and semi-annual scale. Thus, wetter and drier years exerted influence on SWS changes by leading to faster or slower response times in SWS as compared to normal years.

At the daily scale significant correlations in SWS between the two sites were found throughout the entire observation period (Fig. 5 a) without any time shifts (Fig. 5 b, iv) indicating similar diurnal patterns at
both sites.

Dominant climatic deviations in P input and in the onset between the drier and the wetter site were found when correlating the P time series of Dedelow and Selhausen. The P patterns showed significant correlations at the annual scale at the beginning of the observation period in 2014/2015 (Fig. 5 c). A slightly negative phase shift indicated a faster onset of P in Selhausen compared to Dedelow (Fig. 5 d, ii)
that could be attributed to the west wind drift dominating the weather patterns in middle Europe. From 2018 to 2019 a two-year cycle was observed with a positive phase shift (Fig. 5 d, i). Likewise, changed patterns in the SWS could be attributed to carry-over-effects of low P in drought years. In 2021 a significant area in the WCA spectrum was found at the semi-annual scale with a positive phase shift of approximately 12 days (Fig. 5 d, iii) indicating a faster onset of P at the site Dedelow compared to
Selhausen. This corresponds well to the temporal shift between the heavy rainfall events at these two sites





in July 2021. In Dedelow, 174 mm of P were recorded from June 30[th] to July 1[st], 2023. Twelfe days later, Selhausen received 103 mm of P from July 13[th] to July 14[th], 2023. The time shift between the P events was also found at the quarterly scale (Fig. 5 d, iv). This demonstrates the efficiency of WCA to derive information about time shifts that cannot directly be conceived by regular time series analysis.

The shift of the start of vegetation periods towards earlier times of the year over the observation period could be deduced from the WCA spectra of ETa. ETa showed high correlations between Dedelow and Selhausen at the annual scale over the entire period (Fig. 5 e). The correlations were well in phase showing no time shift between the patterns of the two sites (Fig. 5 f). At the semi-annual scale significant correlations occurred throughout the vegetation period (Fig. 5 f, ii). Between 2015 and 2020 the phase

shifts at the semi-annual scale were negative. Since ETa was directly related to the plant development this indicates a faster onset of the vegetation period in Selhausen than in Dedelow with delays of 5 to 15 days as it is found from calculating the onset of the vegetation period from temperature data (Fig. 8). Only in 2021 this shift was inverted to a positive phase shift. As already indicated in the wavelet spectra (Fig. 3 e and f) a highly significant correlation between ETa in Dedelow and Selhausen was found at the daily

scale. The phase shift oscillated around zero hours (Fig. 5 f, iv) indicating similar diurnal patterns for the two sites like it was found for the SWS changes (Fig. 5 b, iv). For drainage our analysis showed for the most years a clear shift between the sites indicating that the rewetting of the same soil started at the wetter site Selhausen earlier in the non-growing season compared to the drier site in Dedelow (Figure C1). Only for the very wet year 2017 a shift towards earlier rewetting in Dedelow is visible. At smaller scales this

is also visible for the extreme precipitation event in 2021 where the precipitation occurred earlier in Dedelow than in Selhausen (Figure C2 and C3).

**3.3 Correlation and time shifts between soil water budget components at each site**

The response time of the ΔSWS to P input was deduced from the WCA spectra between P and ΔSWS. The correlation between P and ΔSWS in Dedelow and Selhausen occurred mainly at smaller scales

corresponding to the return periods of P (Fig. 6 a, c). P and SWS had positive phase shifts across all scales (black arrows pointing upwards) showing that SWS changes were lagging behind P inputs. At a weekly scale this phase shift oscillates around 48 h for Dedelow and Selhausen indicating that approximately 2





days are needed to pass before changes caused by P lead to an increase in SWS (Fig. 6 b, d iv). Similar

temporal delays (0.375 weeks) have been observed for correlations between P and the soil matric potential

in cropland (Yang et al., 2016).



**Figure 6: WCA between precipitation and SWS in Dedelow (a) and Selhausen (c) and time shifts expresses in days (hours) for selected scales in Dedelow (b) and Selhausen (d). For explanation of the plot layout refer to Figure 5.**

Carry-over effects of dry and wet years towards subsequent years were also found when correlating P and

SWS changes. At a two-year scale, significant correlations between P and SWS were identified to occur

between 2017 and 2019 for Dedelow and from 2017 to 2019 and 2020 to 2022 for Selhausen (Fig. 6 b,i;

d,i). This might be attributed to extreme wet (2017 in Dedelow) and dry conditions (2018-2020 in

Dedelow and Selhausen) that were only revealed in significant correlations at scales higher than one year.

Note that the phase shift between the two variables at this scale from 2017 to 2019 was much larger for





Selhausen (~150 days) than for Dedelow (~100 days). A reason for this could be the small $Q_{net}$ in Dedelow
2017: during high amount of rainfall in Dedelow very little water was drained from the lysimeter leading
to greater and probably faster change in SWS in Dedelow as compared to Selhausen. However, the
patterns at the 2-year scale indicate that subsequent extreme years might lead to a carry-over effect in
SWS responses to P that can be derived from deviations in phase shifts in WCA spectra. Groh et al. (2020)

also observed this increased vulnerability of SWS changes in response to droughts. They found that the
SWS after a drought year was not fully restored to its original value after winter when lysimeters were
transferred to a site with a drier and warmer climate. Likewise, at the catchment scale Laaha et al. (2017)
demonstrated that after the severe summer drought in 2015, the SWS has not been recovered. Also,
Boergens et al. (2021) showed that this water deficit event increased for the summer droughts from 2018

to 2019 in comparison to 2015. This might explain why the water deficit was only visible in the WCA
plots at scales > 1 year after 2018 and not before.

Changing time shifts in the correlation between ETa and SWS indicated a shift in the onset of the
vegetation period towards earlier times of the year for the site under a wetter and warmer climate but not
for the drier and colder site. A strong correlation was found between ETa and SWS in Dedelow and

Selhausen at the annual scale. The phase shift was negative indicating that ETa was reacting in response
to SWS changes (Fig. 7 a, c). For Dedelow the phase shift between ETa and SWS stayed constant around
120 days over the entire observation period whereas for Selhausen a decrease in temporal deviations from
136 to 90 days was observed (Fig. 7 b, ii; d, ii). This corresponded to the maximum peak in the global
spectra for ETa and SWS occurring on slightly smaller scales than the annual scale (Fig. 4). Rahmati et

al. (2020) found a similar trend as in Selhausen for grassland lysimeters located in two different climate
regimes. They attributed the decrease in phase shift to a shift of maximum ETa towards earlier times in
the year when at the same time the maximum peak in SWS was delayed over the years. As suggested by
these authors we could demonstrate that an identical phenomenon occurred in cropland. This is most
likely caused due to increasing temperature over the period and the earlier onset of plant decay due to

drought as found by Rahmati et al. (2023). They showed that despite earlier onset of the vegetation period
the length of the growing season has been decreasing to the level of 1981 over Europe due to earlier onset
plant dormancy.







**Figure 7: WCA between evapotranspiration and SWS in Dedelow (a) and Selhausen (c) and time shifts expresses in days (hours) for selected scales in Dedelow (b) and Selhausen (d). For explanation of the plot layout refer to Figure 5.**


However, we did not find such a decreasing phase shift for the soil under the drier and colder climate in Dedelow at the annual scale (Fig. 7 b, ii; d, ii). The phase shift in Dedelow was about 136 h whereas in Selhausen it decreased from 136 to 90 days. One possible reason could be the differing growing season length in Dedelow and Selhausen, which influenced the amount of ETa and SWS. The length of the

vegetation period at both sites was calculated from daily temperature data according to Ernst and Loeper (1976) (Fig. 8) over a thirty-year period from 1992 to 2021 and over the eight-year observation period from 2014 to 2021. The changing length of the vegetation period was calculated for the 30-year period, since the trends were more clearly visible in the longer period in comparison to the shorter eight-year period. For both periods the growing season is longer in Selhausen in comparison to Dedelow as indicated





by the earlier start and later end of the vegetation period in Selhausen. When trying to explain the different time shifts between ETa and SWS in Selhausen and Dedelow, one needs to consider that the soils were relocated according to the space-for-time approach from the drier and colder climate with the shorter growing season in Dedelow to Selhausen, where the growing season is longer and the climate warmer and wetter. Now the decreasing phase shift between ETa and SWS that was observed for Selhausen but

not for Dedelow might exactly indicate the longer growing season in Selhausen that is reflected in earlier maximum peaks of ETa and later maximum peaks in SWS every year. The soils in Dedelow did not experience such a change since they were not subjected to different climatic conditions, whereas the relocated soil in Selhausen had to adapt to the longer vegetation period. With this, an influence of changing climatic conditions on soil water budget parameters of similar soils was detectable.

Interestingly, over the last thirty years the end of the growing season is shifted stronger towards later times in Dedelow as compared to Selhausen (Fig. 8 a). However, the end of the vegetation period for crops is determined by the harvest and not by the actual drop in temperatures. Therefore, the shift in the start of the growing season towards earlier times is more relevant.





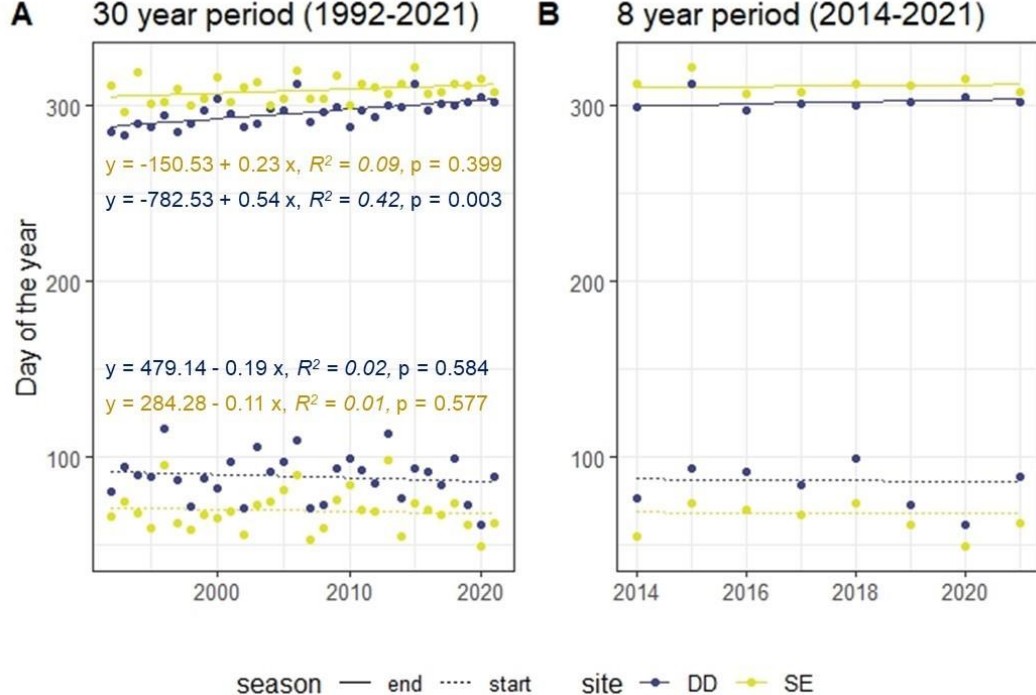

**Figure 8: Variation of beginning and end of the vegetation period in Dedelow (DD) and Selhausen (SE) over a 30-year period from 1992 to 2021 (A) and over the observation period of 8 years from 2014 to 2021 (B). Calculations were executed according to Ernst and Loeper (1976) with hourly temperature data. "End" indicates the days of each year, when the growing season stopped, whereas "start" indicates the days of each year, when growing season started.**

## 4 Conclusions

Soil water storage (SWS) dynamics are important indicators for impacts of environmental changes on the soil-water-atmosphere continuum. Temporal pattern detection and analysis of these changes might help to understand long-term impacts of droughts on plant and crop productivity.

As hypothesized, wavelet coherence analysis (WCA) of soil water balance components from lysimeters with the same soils but under different climatic conditions (drier and colder, wetter and warmer) detected differing temporal patterns with temporal shifts when correlating time series of SWS changes and ETa between both sites. Extreme wet and dry years led to a change in temporal off-set in SWS changes between the two sites. In particular, years with more rainfall led to a faster response in SWS changes than years with less rainfall, as both a lower ETa and an earlier rewetting phase in summer and fall led to a





faster reaction in the SWS changes. This shows how precipitation affects the change in SWS under

different climate conditions.

The impact of droughts on SWS changes was reflected in significant periodic patterns > one year. This implies that dry years led to a carry-over effect in SWS, i.e., the SWS deficit of a dry year affected SWS of the following years. Analysis of longer time series (~30 years) might reveal impact of droughts on even higher scales and long-term carry-over effects.

Most interestingly, the earlier onset of vegetation periods deduced from the correlation between ETa and SWS was only found for the site with a wetter and warmer climate and not for the site under a colder and drier climate. This could be related to the water limitation at the drier site and to changes in the SWS capacity due to the abrupt change in climatic conditions induced by the transfer of the soil monoliths towards the warmer site (space-for-time substitution approach). This could be a first indication that the

change in climatic conditions could have led to changes in the soil water retention capacity as long-term adaption to the new climatic conditions, which could be a topic of future studies. The results of the present study also suggest that long-term time series of SWS changes are important for understanding and quantifying environmental impact of climatic extreme events on soils and cropping systems.



## Code availability

Code will be made available upon request.

## Data availability

Data will be made available upon request.

## Authors contributions

A. Ehrhardt: Conceptualization, Formal analysis, Investigation, Methodology, Software, Validation, Visualization, Writing – original draft preparation, Writing – review & editing

J. Groh: Conceptualization, Data curation, Formal analysis, Investigation, Methodology, Resources, Software, Writing – review & editing

H. Gerke: Conceptualization, Funding acquisition, Methodology, Project administration, Resources, Supervision, Writing – review & editing

## Competing Interests

The authors declare that they have no conflict of interest.

## Acknowledgements

The research was funded by the Leibniz Centre for Agricultural Landscape Research (ZALF), which is a research institution of the Leibniz Association in the legal form of a non-profit registered association. ZALF is financed in equal parts by the Federal Ministry of Food and Agriculture (BMEL) and the Ministry for Science, Research and Culture of the State of Brandenburg (MWFK). The study was also funded by Fachagentur Nachwachsende Rohstoffe e.V. (FNR), grant number 22404117.

We would like to thank Patrizia Ney from Forschungszentrum Jülich for providing the climate data for the study site Selhausen.

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

**Appendix**

**Table A1:** Annual precipitation, actual evapotranspiration (ETa), drainage, upward water flow and change in soil water storage (ΔSWS) for Dedelow (Dd) and Selhausen (Sel) calculated from the lysimeter weights. Data are given in mm a$^{-1}$.





| | Precipitation | | ETa | | Drainage | | Upward Flow | | Δ SWS | |
|---|---|---|---|---|---|---|---|---|---|---|
| | **Dd** | **Sel** | **Dd** | **Sel** | **Dd** | **Sel** | **Dd** | **Sel** | **Dd** | **Sel** |
| **2014** | 676 | 873 | 579 | 676 | 21 | 314 | -36 | -38 | 111 | -79 |
| **2015** | 542 | 744 | 619 | 691 | 78 | 125 | -70 | -87 | -85 | 15 |
| **2016** | 534 | 702 | 556 | 464 | 28 | 271 | -60 | -48 | 10 | 16 |
| **2017** | 872 | 642 | 700 | 601 | 167 | 96 | -42 | -68 | 46 | 13 |
| **2018** | **400** | **534** | 474 | 526 | 109 | 131 | -82 | -77 | -100 | -47 |
| **2019** | 575 | 673 | 572 | 543 | 3 | 155 | -52 | -62 | 52 | 37 |
| **2020** | 498 | **581** | 503 | 475 | 26 | 172 | -31 | -40 | 0 | -26 |
| **2021** | 757 | 768 | 507 | 571 | 188 | 157 | -12 | -61 | 74 | 100 |
| **Sum** | 4854 | 5517 | 4510 | 4547 | 620 | 1421 | -385 | -481 | 108 | 29 |
| **Mean** | 607 | 690 | 564 | 568 | 78 | 178 | -48 | -60 | 14 | 4 |



710 **Figure B1:** Global wavelet spectra Qnet Dedelow and Selhausen

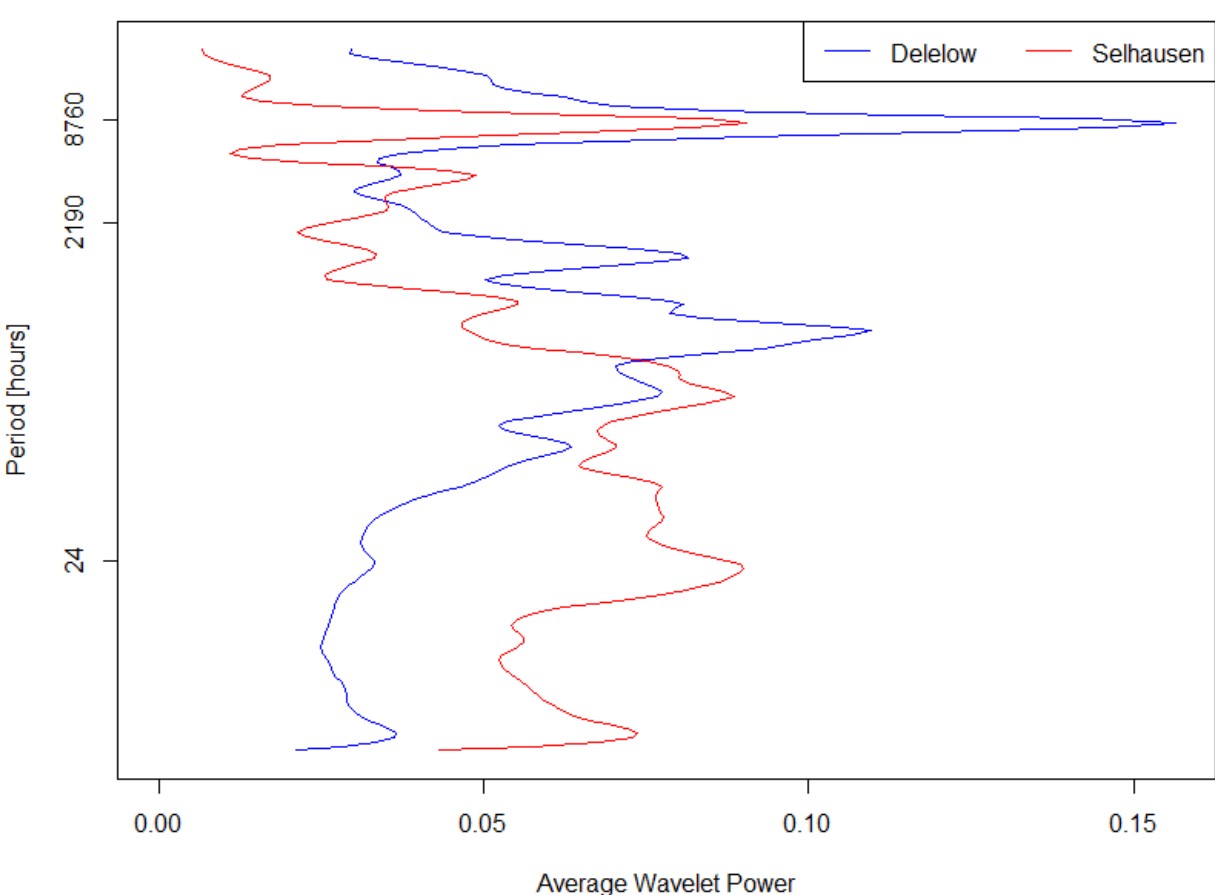



**Figure C1: Wavelet coherency spectrum of drainage in Dedelow and Selhausen**

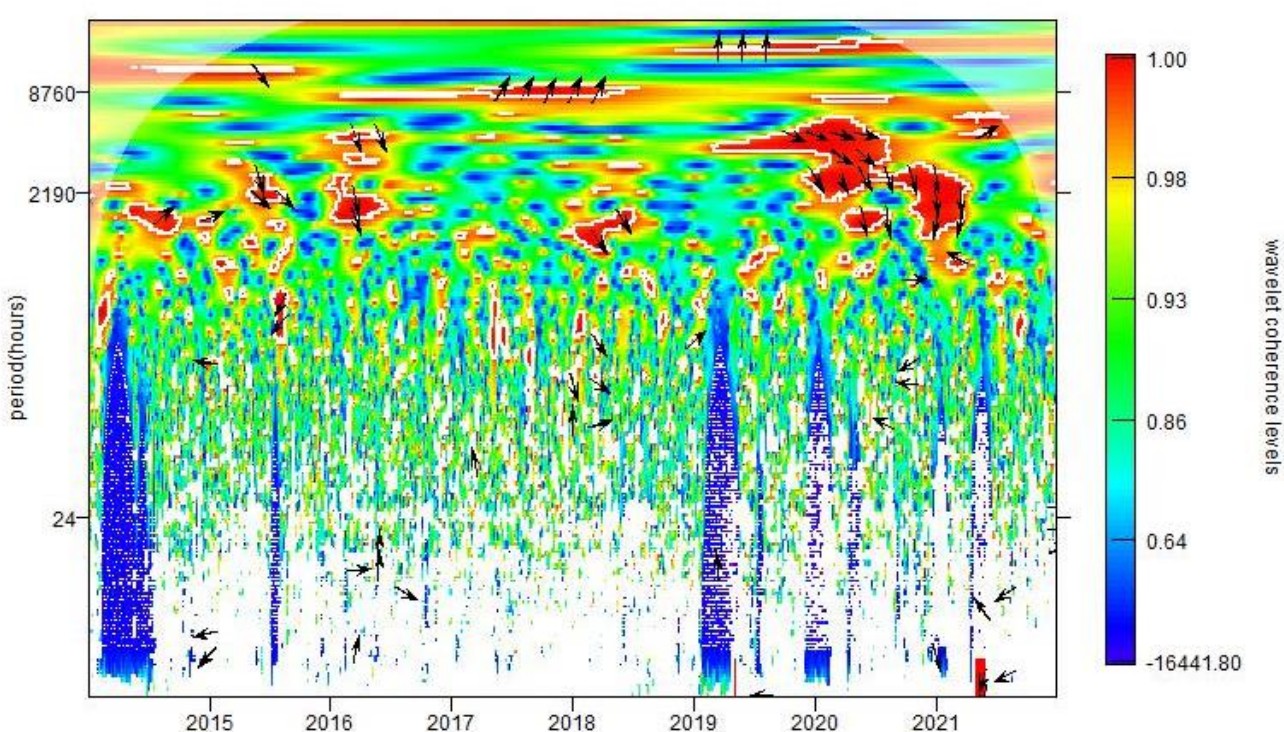

715



**Figure C2: Wavelet coherency spectrum and time shifts between drainage and SWS in Dedelow**



**Figure C3: Wavelet coherency spectrum and time shifts between drainage and SWS in Selhausen**