# Peer review of "Effects of different climatic conditions on soil water storage patterns"

_EGUsphere, 2024_

## Author Comment (AC1)

Dear Reviewer,

thank you very much for your comments to improve our manuscript.

Please see below for a detailed reply to your comments:

- Line 84-112: it's good to see the brief summary of methods used for analyzing time series of soil water. However, in terms of wavelet method, I think it worths mentioning the extension of wavelet coherence from two variables to multiple variables, including multiple wavelet coherence (doi:10.5194/hess-20-3183-2016 ) and partial wavelet coherency (https://doi.org/10.5194/hess-25-321-2021) .
  - Thank you for this suggestion and making us aware of these publications. We will include a paragraph about the multiple wavelet coherence and partial wavelet coherency in the introduction and discussion of our revised manuscript. For the present study, it was sufficient to detect the basic differences in temporal patterns. For future studies, we will definitely consider this approach in our analyses. Especially, partial wavelet coherence is a promising tool to analyze factors (precipitation, Eta) that might explain the variations in SWS.
- Line 113: capitalize "w" in "wavelet" please.
  - We will correct the typo.
- Line 170: I might have missed how did you treat the three replicates when you analyzed the data using wavelet? Did you do wavelet coherency for each lysimeter or for the mean values of the three lysimeters.
  - We used the mean values between the three lysimeters for our further analysis. We will add the following paragraph in the revised version of the manuscript:

    „For the further analysis (wavelet and wavelet coherence analysis) the mean of three replicate lysimeters was calculated for each hour and parameter.

- Line 204-205: I would detail the exact depth of each horizon for each lysimeter. How did the variations in the thickness of various horizons below the Ap horizon affect the SWS and associated correlations with climate (e.g., P, and Eta)?
  - We will include a table with the exact depths of each horizon in the appendix. As you can see in Fig. 2 the variation between the different lysimeters is very small as well as the variation in horizon depth between the lysimeters. Therefore, we decided to include only the plots of the mean values between the different lysimeters for each site in the manuscript.

| Horizon DD-01 | Depth [cm] DD-01 | Horizon DD-01 | Depth [cm] DD-03 | Horizon DD-05 | Depth [cm] DD-05 | Horizon SE-41 | Depth [cm] SE-41 | Horizon SE-45 | Depth [cm] SE-45 | Horizon SE-46 | Depth [cm] SE-46 |
|---|---|---|---|---|---|---|---|---|---|---|---|
| Ap | 0-30 | Ap | 0-35 | Ap | 0-30 | Ap | 0-30 | Ap | 0-37 | Ap | 0-35 |
| Al+Bt | 30-42 | Bt | 35-75 | Bt | 30-65- | Bt | 30-55 | Bt | 37-75 | Bt | 35-75 |
| Bt | 42-80 | elCcv1 | 75-115 | elCcv1 | 65-115 | elCcv | 55-100 | elCcv | 75-150 | Bvt | 75-84 |
| elCcv | 80-150 | elCcv2 | 115-150 | elCcv2 | 115-150 | elCv | 100-150 | | | elCcv | 84-105 |
| | | | | | | | | | | elCv | 105-150 |

- Line 208: why not keep exact the same. How can you exclude that the different crops in 2014 would not affect the associated relationships?
  - You absolutely right that it would have been more reasonable to plant the same crops at both sites in 2014. Unfortunately, the effect cannot be excluded that the different crops might have on the SWS. However, we decided to keep the year 2014 in our analysis to extend the observation period at the beginning in order to be better informed on initial differences. The different crops were planted in the beginning and not in the middle of the time period, effects were assumed to be minimal.
  - We just received note that in 2015/16 winter wheat instead of winter barley was planted in Dedelow. However, since both crops are winter cereals we expect only minor deviances.
- Line 245: if you are interested in the real correlation between two variables, partial wavelet coherency mentioned above may be a better option. This at least can be discussed in the conclusion.
  - Thanks for the suggestion. As you mentioned, partial wavelet coherency (PWC) might be a better tool to analyse the correlations between soil water storage, precipitation and actual evapotranspiration. According to the publication you mentioned PWC is especially useful when dealing with variables that might be dependent on other variables. By applying PWC this effect of other variables can be excluded. We will include a paragraph of this advanced methodology in our discussion and conclusion.
  - Note that PWC is currently implemented in the commercial software Matlab, but not implemented in free software R.
- Line 296: I don't think that band is green, more like bright sky blue
  - We will change the colour description.
- Line 301, 304, 305: please specify which smaller scales
  - With smaller scales we meant the scales from semi-annual to monthly scales. We will specify that in the revised version of the manuscript.
- Line 340: can't see the small peak in Fig 4b. Do you mean Fig 4d?
  - We meant Figure 4d. Sorry for the confusion. We will correct the mistake.
- Line 350: I did not see the description of rainfall pattern. It shows no annual cycle but big peak at a few hours' time scales, and this is more obvious at the drier site. Can you please add this result in?
  - Thank you for the remark. We mentioned the rainfall pattern in line 340 but as you suggested, we could enhance the description more. We will include the results in the edited manuscript.
- Line 396: Twelve
  - We will exchange „12" for „twelve".
- Line 451-454: can you please explain how ETa responds to the SWS changes after more than 100 days? ETa should not respond to SWS change in a very short time? I know this is related to different time scale, but it seems really hard to understand from the hydrological process point of view. You may need to clarify here.
  - Why is there such a temporal scale in the relation between SWS and ETa? From a hydrological perspective, SWS and ETa are of course always related from shorter to longer times. The time delay in the relation between ETa and changes in SWS at shorter times (hourly, daily, etc) are, however, stronger affected by other water balance components. However, the time scale we are looking at is the annual scale so the variations we are observing here are more related to seasonal fluctuations than small-scale daily fluctuations. So at a seasonal scale the SWS is decreasing around 90 days earlier than the ETa,

which could mean that the decrease in ETa could be buffered by taking up water from deeper layers of the soil. So the SWS will decrease but not the ETa. This shows the importance of SWS as a variable for crop productivity.

- o We will explain this fact more detailed in the revised manuscript.

- Line 467: 136 h or day?
  - o We meant „days" and will correct the mistake in the manuscript.

---

## Author Response (AR1)

**Reply Reviewer 1**

Dear Reviewer,

thank you very much for your comments to improve our manuscript.

Please see below for a detailed reply to your comments:

- Line 84-112: it's good to see the brief summary of methods used for analyzing time series of soil water. However, in terms of wavelet method, I think it worths mentioning the extension of wavelet coherence from two variables to multiple variables, including multiple wavelet coherence (doi:10.5194/hess-20-3183-2016 ) and partial wavelet coherency (https://doi.org/10.5194/hess-25-321-2021) .
  - Thank you for this suggestion and making us aware of these publications. We will include a paragraph about the multiple wavelet coherence and partial wavelet coherency in the introduction (ll. 143-150) and conclusion (597-598) of our revised manuscript. For the present study, it was sufficient to detect the basic differences in temporal patterns. For future studies, we will definitely consider this approach in our analyses. Especially, partial wavelet coherence is a promising tool to analyze factors (precipitation, ETa) that might explain the variations in SWS.
- Line 113: capitalize "w" in "wavelet" please.
  - We will correct the typo (l. 128).
- Line 170: I might have missed how did you treat the three replicates when you analyzed the data using wavelet? Did you do wavelet coherency for each lysimeter or for the mean values of the three lysimeters.
  - We used the mean values between the three lysimeters for our further analysis. We will add the following paragraph in the revised version of the manuscript:

    „For the further analysis (wavelet and wavelet coherence analysis) the mean of three replicate lysimeters was calculated for each hour and parameter." (ll. 279-280)

- Line 204-205: I would detail the exact depth of each horizon for each lysimeter. How did the variations in the thickness of various horizons below the Ap horizon affect the SWS and associated correlations with climate (e.g., P, and Eta)?
  - We depicted the depth of each horizon for each single lysimeter below. As you can see in Fig. 2 the variation between the different lysimeters is very small as well as the variation in horizon depth between the lysimeters. Therefore, we decided to include only the plots of the mean values between the different lysimeters for each site in the manuscript. The data printed here is already published. We refer to this in the manuscript in ll. 238-241.

| Horizon DD-01 | Depth [cm] DD-01 | Horizon DD-01 | Depth [cm] DD-03 | Horizon DD-05 | Depth [cm] DD-05 | Horizon SE-41 | Depth [cm] SE-41 | Horizon SE-45 | Depth [cm] SE-45 | Horizon SE-46 | Depth [cm] SE-46 |
|---|---|---|---|---|---|---|---|---|---|---|---|
| Ap | 0-30 | Ap | 0-35 | Ap | 0-30 | Ap | 0-30 | Ap | 0-37 | Ap | 0-35 |
| Al+Bt | 30-42 | Bt | 35-75 | Bt | 30-65- | Bt | 30-55 | Bt | 37-75 | Bt | 35-75 |
| Bt | 42-80 | elCcv1 | 75-115 | elCcv1 | 65-115 | elCcv | 55-100 | elCcv | 75-150 | Bvt | 75-84 |
| elCcv | 80-150 | elCcv2 | 115-150 | elCcv2 | 115-150 | elCv | 100-150 | | | elCcv | 84-105 |
| | | | | | | | | | | elCv | 105-150 |

- Line 208: why not keep exact the same. How can you exclude that the different crops in 2014 would not affect the associated relationships?
  - You absolutely right that it would have been more reasonable to plant the same crops at both sites in 2014. Unfortunately, the effect cannot be excluded that the different crops might have on the SWS. However, we decided to keep the year 2014 in our analysis to extend the observation period at the beginning in order to be better informed on initial differences. The different crops were planted in the beginning and not in the middle of the time period, effects were assumed to be minimal. We added this in the manuscript in ll. 247-250.
  - We just received note that in 2015/16 winter wheat instead of winter barley was planted in Dedelow (Tab. 2). However, since both crops are winter cereals we expect only minor deviances.
- Line 245: if you are interested in the real correlation between two variables, partial wavelet coherency mentioned above may be a better option. This at least can be discussed in the conclusion.
  - Thanks for the suggestion. As you mentioned, partial wavelet coherency (PWC) might be a better tool to analyse the correlations between soil water storage, precipitation and actual evapotranspiration. According to the publication you mentioned PWC is especially useful when dealing with variables that might be dependent on other variables. By applying PWC this effect of other variables can be excluded. We will mention advanced methodology in our conclusion (ll. 597-598).
  - Note that PWC is currently implemented in the commercial software Matlab, but not implemented in free software R.
- Line 296: I don't think that band is green, more like bright sky blue
  - We will change the colour description (ll. 345-346).
- Line 301, 304, 305: please specify which smaller scales
  - With smaller scales we meant the scales from semi-annual to monthly scales. We will specify that in the revised version of the manuscript (ll. 351-355).
- Line 340: can't see the small peak in Fig 4b. Do you mean Fig 4d?
  - We meant Figure 4d. Sorry for the confusion. We will correct the mistake (l. 392).
- Line 350: I did not see the description of rainfall pattern. It shows no annual cycle but big peak at a few hours' time scales, and this is more obvious at the drier site. Can you please add this result in?
  - Thank you for the remark. We will include the results in the edited manuscript.

"For P no annual pattern was found in the global wavelet spectra but at a periodicity of approximately hours, a peak was observed in both spectra (Fig. 4 b). This peak was more pronounced for the drier site in Dedelow, however, the global wavelet power was much smaller in comparison to SWS and $ET_a$" (ll. 394-396)

- Line 396: Twelve
  - We will correct the typo (l. 452).
- Line 451-454: can you please explain how ETa responds to the SWS changes after more than 100 days? ETa should not respond to SWS change in a very short time? I know this is related to different time scale, but it seems really hard to understand from the hydrological process point of view. You may need to clarify here.
  - Why is there such a temporal scale in the relation between SWS and ETa? From a hydrological perspective, SWS and ETa are of course always related from shorter to longer times. The time delay in the relation between ETa and changes in SWS at shorter times (hourly, daily, etc) are, however, stronger affected by other water balance components. However, the time scale we are looking at is the annual scale so the variations we are observing here are more related to seasonal fluctuations than small-scale daily fluctuations. So at a seasonal scale the SWS is decreasing around 90 days earlier than the ETa, which could mean that the decrease in ETa could be buffered by taking up water from deeper layers of the soil. So the SWS will decrease but not the ETa. This shows the importance of SWS as a variable for crop productivity.
  - We will explain this fact more detailed in the revised manuscript (ll.526-534).

- Line 467: 136 h or day?
  - We meant „days" and will correct the mistake in the manuscript (l. 539).

**Reply Reviewer 2**

Dear Reviewer,

thank you very much for your comments that helped to improve our manuscript.

Please see below for a detailed reply to your comments:

- This part can be better organized. For example, you mentioned that "Pattern identification and quantification of these variations remains difficult", you mean the variations in SWS? if so, why not just analyze the measured SWS? Why you believe "these patterns can be revealed by applying wavelet analysis"? What inspired you to conduct such an analysis? Please clarify.
    - Yes, good question, measured values can be compared as well. But patterns can give more generalized results/ information on the differences between two sites. And we assumed that we could see a transition of the patterns from those at the original to those at the new site. Also, the benefit of WCA is the possibility to analyse temporal correlations of SWS at every point in time of the time series instead of simply looking at correlations coefficients that are averaged across the time series (e.g. Pearson). Particularly, wavelet analysis decomposes a time series into several components each accounting for a certain frequency band by comparing the signal with a set of wavelet functions of known frequency. Additionally, when analysing patterns between two time series is to find possible correlations in these often nonstationary datasets (Ritter et al., 2009) to identify differences and similarities. Wavelet coherency analysis (WCA) can reveal such similarity between two signals that might have been overlooked by traditional correlation analysis (Grinsted et al., 2004). For example, if two time series contain similar frequencies but are only shifted in time against each other Pearson correlation indicates only little similarity between the signals in contrast to WCA (Bravo et al., 2019)."
    - We will clarify this in the revised manuscript by editing the beginning of the abstract and later in the introduction (see comment 3):
    - *"Pattern identification and quantification of these variations in SWS remains difficult due to the non-linear behaviour of SWS changes over time. Wavelet analysis (WA) provides a tool to efficiently visualize and quantify these patterns by transferring the time series from time to frequency-domain. We applied WA to ..."* (ll. 18-20).
- Also, you concluded in Abstract that "wet and dry years exerted influence on SWS changes by leading to faster or slower response times of SWS changes to precipitation in respect to normal years." But why? does that caused by extreme precipitation events? why you believe "Long-term observations (>30 years) might reveal similar time shifts for a drier climate" ?
    - We found that wet and dry years interrupt the annual pattern observed in the wavelet spectra for both sites. This might be caused by extreme events as you already indicated. The disruptions of this annual cycle will also affect temporal variations in the correlation between the drier and the wetter site. We will add this explanation in the abstract (ll. 29-30).
    - We found a decrease in phase shift between ETa and SWS at the wetter and warmer site that was not observed for the drier and colder site. Assuming that the climate at the colder and drier site will also change due to climate change, we expect that the time shifts at the drier site will also be affected. This can only be assessed, if we analyze longer time series. However, this is a

speculation that might be misinterpreted so we decided that we will remove this statement from the revised manuscript (ll-35-37).

- I found that the logic in some paragraphs is hard to follow, there are too much plain concepts and descriptions. For example, the paragraph talking about the methods of deriving reoccurring patterns in time series of SWS, all of these methods were fairly detailed in other researches using time series analysis. I believe the advantages of wavelet coherency analysis and the reason for taking the method in this study should be better highlighted.
  - o That is a good suggestion. We will shorten the paragraph with the other methods and will highlight the advantages of WCA more, e.g. like this:
  - o "*To analyse these dynamics and derive reoccurring patterns in time series of SWS, a variety of methods including principal component analysis (PCA), empirical orthogonal functions (EOF), wavelet transform, unsupervised learning like self-organizing maps (SOM), empirical mode decomposition (EMD) have been applied (Vereecken et al., 2016). However, these approaches do not allow to localize these patterns in time as it could be done with a wavelet analysis. Especially, it is not possible to determine whether annual or daily cycles within a signal are occurring over the entire period or if these patterns are interrupted in time. Wavelet analysis provides such a tool by decomposing a time series into several components each accounting for a certain frequency band by comparing the signal with a set of wavelet functions of known frequency. [...]. Additionally, when analysing patterns between two time series is to find possible correlations in these often nonstationary datasets (Ritter et al., 2009) to identify differences and similarities. Wavelet coherency analysis (WCA) can reveal such similarity between two signals that might have been overlooked by traditional correlation analysis (Grinsted et al., 2004). For example, if two time series contain similar frequencies but are only shifted in time against each other Pearson correlation indicates only little similarity between the signals in contrast to WCA (Bravo et al., 2019).*" (ll.90-98 and 119-127)
- In line 77, you mentioned "the effect of a change in climatic conditions on SWS has scarcely been reported to date." But in lines 57-65, several papers were cited, please explain more.
  - o We meant the effect on soil water storage **patterns** has scarcely been reported to date. The mentioned studies reported distinct effects of climate extremes on soil water storage in single years. However, our aim is to identify long-term patterns that support the hypothesis that there are overall changes in SWS induced by a change of climatic conditions. We will clarify this in the revised manuscript.
  - o "*However, the effects of changing climatic conditions on temporal patterns in SWS time series have not been widely reported. Identification of such patterns might help to derive the impact of climate change on SWS as an important component of the ecosystem water balance.*" (ll. 81-84).
- Line 113, capitalize the first letter.
  - o We will correct the typo ((l. 128).
- Lines 128-129. The authors mentioned: "When analyzing the effect of climate variability on SWS it is plausible to compare time series of similar soils under different climatic conditions", why similar soils? In my opinion, soil is also part of results in a given climatic condition, so what is the practical meaning of this experiment? Needing further explain.

- o You are totally right that soil formation is also a result of different climatic conditions. However, we assume that the soil changes are slower than the changes in the water balance components, especially with regard to human-induced climate change, which is much faster than climate variablity. Our aim was to derive what impact the change in climate today would do to a soil. To put it bluntly, we would like to simulate the effects of climate change on soils by comparing soils that remain in the climate of their formation to the same soil relocated to a different climate. In other words, if the soil and crop rotation remain similar and only the climate differs, than deviations in SWS pattern between two soils must be attributed to climate. We will clarify this in the revised manuscript.
  - o *"When analysing the effect of climate variability on SWS it is plausible to compare time series of similar soils under different climatic conditions (i.e., space-for-time substitution approach, e.g., Groh et al., 2020a). If deviations in soil type and crop rotation can be excluded, deviations in SWS patterns between the two places must be attributed to different climatic conditions. The hypothesis is that if there are no differences in SWS patterns between the two sites, climatic conditions do not affect SWS."* (ll. 151-155)
- Lines 157-158. you hypothesized that "similar to grassland soils the phase shift between ETa and SWS is smaller under drier as compared to wetter conditions". but why? As we know the crop land has totally different hydrological characteristics from grasslands, why you believe the SWS variation patterns of them are similar?
  - o You are right that cropland and grassland have different hydrological characteristics. While editing the structure of the manuscript with the comments of you and your fellow reviewers we realized that this hypothesis does not fit the overall message of the paper. At that point we would rather like to highlight the point you mentioned above "what is the practical meaning of the experiment" and connect it to the following hypothesis: *We want to analyse, if SWS patterns can be assumed to be independent of the site-specific climatic conditions and thus be assumed to be entirely dependent on the soil conditions. We hypothesize that there is no variation in SWS of the similarly managed arable soils at the two sites if SWS patterns are independent of the climatic conditions.* (ll. 183-188)
- Figure 1, scales of the two enlarged maps are obviously different, and unify scales are recommended, latitude and longitude also need to be included. Besides, the text was too small and not easy to read.
  - o We will add scales to the figure and the text will be enlarged.
- Explanatory text in Figure 1 was not accurate enough (only mentioned the average monthly precipitation (P) sums and average monthly temperature) and thus need to further clarified.
  - o Thanks, we forgot to explain the gradient in temperature and precipitation between the two places. We will change the caption to:
  - o *"Average monthly precipitation (P) sums and average monthly temperature in Selhausen (left, located in the west of Germany) and in Dedelow (right, located in the northeast of Germany) (between 1991 and 2022). Red and blue arrow indicate the gradient in temperature and precipitation. Dedelow receives on average 197 mm less precipitation than Selhausen and is on average 2.5 °C cooler than the site in western Germany."* (ll.217-221)
- How about the influence of amount of precipitation during the vegetation period as your record in Table 2?

- o We are not entirely sure to which line this comment refers to and which variable it refers to. However, we used the information of the vegetation period to explain differences in ΔSWS time series at both sites. For example: The vegetation period precipitation amount was used to explain difference in ΔSWS time series for Dedelow (ll. 357-358) and when comparing both time series of ΔSWS in Dedelow and Selhausen (ll. 426-428).
- Section 2.2, as you said in line 195, "the average monthly temperatures and precipitation were obtained from automated weather stations", and in line 218, "Missing data were gap-filled on aggregated hourly basis within the post-processing scheme", considering the 1-min resolution collection in lysimeters, how did you ensure the accuracy of precipitation data post-processed ? Are there any uncertainties from this processing?
  - o The long-term weather observations (1991-2022) were obtain from a close by station at Selhausen and Dedelow, as the lysimeter were established in 2010. (ll.257-259)
  - o Regarding the second question: The precipitation data were obtained from weight changes of the lysimeter and parallel observations from other lysimeters, and in a first step, linear regression models were used to fill in the time series first on a 10-minute and then on an hourly time scale. The last missing values were filled in with hourly precipitation data from the lysimeter weather station. A linear regression model was also used here. A detailed comparison between precipitation data from Lysimeter and standard rain gauges can be found in the following publication (Schnepper et al. 2023). (ll. 264-269)
- Equation 1, P was from automated weather stations, Qnet from lysimeters, but where did ETa come from? You didn't give explicit data source.
  - o P and Eta was calculated from the lysimeters, as proposed by e.g., Schrader et al. 2013 or Schneider et al. 2021,. We did not use externe rain gauges for obtaining hourly P values. As previously mentioned, P values from standard rain gauges underestimate the amount of P partially largely (see more details in Schnepper et al. 2023). We will clarify this in the revised manuscript. (ll.257-259)
- Figure 2, legend "dd" and "sel", while "DD" and "SE" in Figure 8, it is better to be consistent.
  - o We will change the legend in figure 8.
- In line 294, "They related these fluctuations to seasonal variations due to water consumption by plants (transpiration) and soil evaporation." Is that the same reason for the changes reported in your study?
  - o Yes, we assume that these seasonal variations can be attributed to transpiration. We will add this explanation in the revised manuscript. (ll. 343 – 344)
- Line 332, "the periodicities at the daily scale were significant throughout the vegetation period at both sites", in the current drawing forms, the daily scale periodicities were not obvious to obtain.
  - o The significance of the periodicities at the daily scale is indicated by the black lines at the 24-h-scale. Since at smaller scales the white rim indicating significant areas in the wavelet plots is rather omnipresent, the software indicates the average significant periodicities by black lines. This is however not mentioned in the description of Figure 3 and we will add this in the revised manuscript. (ll. 365-366)
- Line 339, "In contrast to Dedelow, a small peak around a period of approximately 16500 hours was found in Selhausen", similar to comment 8, not obviously.

- - This was a typo. We refer here to Figure 4d, the close-up of Figure 4a. There the peak around 16500 hours in Selhausen should be fairly visible. We will correct that in the revised manuscript. (l. 392)
- As you said in line 486, "the end of the vegetation period for crops is determined by the harvest and not by the actual drop in temperatures", while calculations were executed according to Ernst and Loeper (1976) with hourly temperature data in Figure 8, please clarify.
  - With the analysis by Ernst & Loeper we tried to analyse, if there is an effect of the vegetation period length that might explain the deviations in SWS changes between the places. E.g., if the vegetation period started earlier every year at one place but not the other site than this might explain the differences found. However, the differences for the sites were only obvious for the end of the vegetation period, which is in our case not relevant, since we analyse cropland. Thus, the difference in the end of the vegetation period cannot be used to explain the found differences SWS patterns between Dedelow and Selhausen. We will clarify this in the revised manuscript. (ll. 559-564)
- Most importantly, the study areas in your paper were located in Selhausen (51°52'7''N, 6°26'58''E) and Dedelow (53°23'2''N, 13°47'11''E), is the climatic discrepancy between them significant enough to call them "different climatic conditions" as section 3.1 and your title ?
  - Yes, one might argue, if this difference might already be called "different climatic conditions". We have certainly not relocated our lysimeters to two different climate zones. However, by analysis of precipitation data and temperature data as given in Figure 1 there is a distinct climatic gradient between the two places, that might represent climatic changes expected due to climate change (at least for the temperature). In addition we will also include other important variable to clarify the different climatic conditions at both site: By relocating the lysimeters from Dedelow to Selhausen the soils were subjected to a higher annual average air temperature (+2.5°C), rainfall (197 mm a$^{-1}$) as well as a lower potential Evapotranspiration (-122 mm a$^{-1}$) and a slightly lower wind speed (0.3 m s$^{-1}$). We will add the difference in wind speed and ET0 in the manuscript. (ll. 231-233)
- The tables need to be better organized.
  - We will reorganize the tables so that table headers are better separated from the table body.
- Discussion is not sufficient in section 3.2. The reason for time shifts is lacking, and the implication of these results needs be illustrated better.
  - We have described probable reasons for the time shifts in section 3.2: For SWS we attribute these time shifts to carry-over effects of extreme years and to the impact of extreme years. In the precipitation spectrum the time shifts indicate deviations found to the different longitude of both locations and the pattern that is expected because of the European West wind drift. When we consider ETa, the earlier onset of the vegetation period might be the reason for the observed time shift. These results imply that climatic conditions indeed have a distinct effect on SWS patterns, that are especially found in extreme years. As the climate is about to become more extreme, e.g. as suggested by Rahmstorf et al. (2023) by the weakening of the gulf stream in northern Europe, these patterns might persist over the years. Temporal changes in SWS increase over winter time and decrease over summer time will affect crop production or the infiltration capacity of soils during extreme events. Crops might need to be planted earlier but also harvested earlier due to an earlier water deficit in summer, as it is already suggested by German agencies (. We will improve the

discussion section to make this important finding more clear in the section and also provide more clearly the implications of the results in the Conclusion section. (ll. 455-457, ll. 475-479)

References:

Rahmstorf, S.: Is the atlantic overturning circulation approaching a tipping point?, Ocenanography, doi:10.5670/oceanog.2024.501., 2024.

Ritter, A., Regalado, C. M., and Muñoz-Carpena, R.: Temporal Common Trends of Topsoil Water Dynamics in a Humid Subtropical Forest Watershed, Vadose Zone Journal, 8, 437–449, doi:10.2136/vzj2008.0054, 2009.

Schnepper, T., Groh, J., Gerke, H. H., Reichert, B., and Pütz, T.: Evaluation of precipitation measurement methods using data from a precision lysimeter network, Hydrol. Earth Syst. Sci., 27, 3265–3292, https://doi.org/10.5194/hess-27-3265-2023, 2023.

Schrader, F., Durner, W., Fank, J., Gebler, S., Pütz, T., Hannes, M., and Wollschläger, U.: Estimating Precipitation and Actual Evapotranspiration from Precision Lysimeter Measurements, Procedia Environmental Sciences, 19, 543–552, doi:10.1016/j.proenv.2013.06.061, 2013.

Guddat, C.; Schwabe, I.: Thüringer Pflanzenbau im Klimawandel; Thüringer Landesanstalt für Landwirtschaft (2012): https://www.tlllr.de/www/daten/agraroekologie/klima/klimawandel/pflanzenbau_klimawandel_thu eringen.pdf

---

## Author Response (AR2)

Dear Reviewer,

thank you very much for your comments to improve our manuscript.

Please see below for a detailed reply to your comments:

- TITLE: Also if it could be quite "appealing", it can result misleading; so I suggest rephrasing for example "Effects of different climatic conditions on soil water storage patterns
  - We changed the title according to your suggestion
- ABSTRACT: "This year and reflects the direct impact of changing climate on soil water budget parameters." Unclear as it seems related to the effects of global warming and not to the comparison between two different conditions, please remove or rephrase
  - We changed "changing climate" to "different climatic conditions", so the phrase does not refer to global warming, as you suggested.
- L46: "The SWS is the residual between in-flux and out-flux components of the soil water balance;" it could be trivial or incorrect (as also bottom drainage can occur, I suggest removing)
  - We removed the sentence as you suggested
- L117: I suppose that the meaning of the sentence could result unclear after the track change mode
  - The long sentence is rather confusing, especially with "track changes mode" turned on. Thus, we split the sentence into two sentences, so our message would become more evident: "Thus, the dominant frequencies of a time series can be derived with WA for each moment in time. In contrast, Fourier analysis calculates only the dominant frequency across the entire time series (Torrence and Compo, 1998)." (ll. 104-106)
- L92-155: the paragraphs are hard to read as they anticipate the paragraph §2.3 where many concepts are introduced. It is a general comment as I know that, at this stage, restructuring the sections could be hard
  - Thank you for the comment. We understand that the section about Wavelet analysis and its application is rather extensive and that some parts are better placed in the method section 2.3. However, since this section describes concepts and not distinct methodological facts that are needed to reproduce our research, we would prefer leaving the section as it is. One could argue that some concepts like PWC and MWC could be removed since they are not applied and thus directly relevant to the paper. On the other hand, we were asked to include such a section during an earlier review stage.
- L205: the time span (2014-2021) is too short to identify robust variations in thermometric and rainfall regimes; so it could be better to report how, during the observed period, these were the variations or considering longer time span to estimate actual differences in climatology
  - You are right: the study period is too short to report robust variations in temperature and rainfall. That is why we included the longterm climatic differences (1991-2022) (Figure 1, new manuscript), that also support the short term differences we see in the data. Reasons for that might be the more maritime climate in Selhausen compared to Dedelow. We emphasized this in the text by adding: "Thus, within these eight years, the lysimeters from

Dedelow were exposed to wetter and warmer weather conditions caused by the more oceanic climate in Selhausen as compared the more continental one in Dedelow. Also, considering the two sites with the different climatic conditions, the weather was characterized by extreme rainfall events (2017), relatively wet (2017, 2021) and dry (2018) periods within the observation period. Compared with the longer-term periods, these extremes seem to be exceptional." (ll. 190-196)
-
- Figure 1: If I have correctly understood the meaning of the values in the graphs, that related to Dedelow should be corrected (now they report the same values)
  - We corrected the wrong numbers in the graph of Dedelow and checked the manuscript again. The numbers in the text were correct.
- L232: how do you compute potential evapotranspiration?
  - While checking our metadata for the calculation of the ETP in Dedelow and Selhausen, we realized that in Selhausen the grass reference evapotranspiration (ET0) was calculated (according to Penman-Monteith) and in Dedelow the potential evapotranspiration according to Wendling et al. (1991). Since these are different types of evapotranspiration that should not be confused. The difference in ETP or ET0 between our two sites was only used to characterise the gradient in climate between Dedelow and Selhausen and is not important for the interpretation of our results we decided to remove this value from the manuscript.
- Table 1: are information about soil hydraulic properties available? E.g. soil-water characteristic curves or hydraulic conductivity values?
  - Yes, there are data on soil hydraulic properties available. They are published in Herbrich & Gerke (2017) and Rieckh et al. (2012) as well in supplementary table S8 of Groh et al. (2022). We added the reference that soil hydraulic properties can be found in these publications (header of Table 1)
- L580: typo for "actual"
  - We corrected the typo.

Literature:

Groh, J., Diamantopoulos, E., Duan, X., Ewert, F., Heinlein, F., Herbst, M., Holbak, M., Kamali, B., Kersebaum, K.-C., Kuhnert, M., Nendel, C., Priesack, E., Steidl, J., Sommer, M., Pütz, T., Vanderborght, J., Vereecken, H., Wallor, E., Weber, T. K. D., Wegehenkel, M., Weihermüller, L., and Gerke, H. H.: Same soil, different climate: Crop model intercomparison on translocated lysimeters, Vadose Zone Journal, 21, 303, doi:10.1002/vzj2.20202, 2022.

Herbrich, M. and Gerke, H. H.: Scales of Water Retention Dynamics Observed in Eroded Luvisols from an Arable Postglacial Soil Landscape, Vadose Zone Journal, 16, 1–17, doi:10.2136/vzj2017.01.0003, 2017.

Rieckh, H., Gerke, H. H., & Sommer, M. (2012). Hydraulic properties of characteristic horizons depending on relief position and structure in a hummocky glacial soil landscape. Soil and Tillage Research, 125, 123–131. https://doi.org/10.1016/j.still.2012.07.004

Wendling, U., Schellin, H.G., and Thomä, M.: Bereitstellung von täglichen Informationen zum Wasserhaushalt des Bodens für die Zwecke der agrarmeteorologischen Beratung, Z. Meteorol., 41, 468–475, 1991.